# Modelling single-cell RNA-seq trajectories on a flat statistical manifold

**Alessandro Palma**[*]
Helmholtz Munich
alessandro.palma@helmholtz-munich.de

**Sergei Rybakov**[*]
Helmholtz Munich
sergei.rybakov@helmholtz-munich.de

**Leon Hetzel**[*]
Helmholtz Munich
leon.hetzel@helmholtz-munich.de

**Fabian Theis**
Helmholtz Munich
fabian.theis@helmholtz-munich.de

## Abstract

Optimal transport has demonstrated remarkable potential in the field of single-cell biology, addressing relevant tasks such as trajectory modelling and perturbation effect prediction. However, the standard formulation of optimal transport assumes Euclidean geometry in the representation space, which may not hold in traditional single-cell embedding methods based on Variational Autoencoders. In this study, we introduce a novel approach for matching the latent dynamics learnt by Euclidean optimal transport with geodesic trajectories in the decoded space. We achieve this by implementing a "flattening" regularisation derived from the pullback metric of a Negative Binomial statistical manifold. The method ensures alignment between the latent space of a discrete Variational Autoencoder modelling single-cell data and Euclidean space, thereby improving compatibility with optimal transport. Our results in four biological settings demonstrate that these constraints enhance the reconstruction of cellular trajectories and velocity fields. We believe that our versatile approach holds promise for advancing single-cell representation learning and temporal modelling.

## 1 Introduction

Temporal modelling is a widely-explored branch of machine learning with successful implications in applied sciences [Heaton et al., 2016, Hewage et al., 2020]. In single-cell biology, modelling a system's evolution through time encompasses tracing how a cell state measured by single-cell RNA-seq (scRNA-seq) develops across a plethora of biological processes, ranging from development to disease progression [Ding et al., 2022]. However, traditional single-cell sequencing is a destructive practice, meaning that the same cell is measured only once and cannot be explicitly tracked through time [Haque et al., 2017]. This aspect poses challenges, in that measurements collected from time-resolved scRNA-seq do not contain direct information about individual cell state progression. Hence, learning about the system's dynamics requires the reconstruction of cellular trajectories from unpaired single-cell distributions sampled across experimental time.

Several methods have proven successful at reconstructing cellular trajectories from snapshots of cells collected over time [Haghverdi et al., 2016, Jürges et al., 2018, Fischer et al., 2019]. Many of them rely on the concept of Optimal Transport (OT) [Peyré and Cuturi, 2017], which matches cell distributions across consecutive time points. OT has been readily employed to interpolate single-cell transcriptomics snapshots and study the cell population dynamics [Schiebinger et al., 2019, Tong

---

[*]Equal contribution.

NeurIPS 2023 AI for Science Workshop.

et al., 2020]. However, single-cell data holds properties that make the application of OT in the gene space challenging. More specifically, the curse of dimensionality complicates OT applications to cells, as OT relies on computing distances between sparse transcription vectors measuring thousands of genes at the same time. To circumvent this issue, the majority of the current methods resort to applying OT to linear projections of data into a low dimensional space via Principal Component Analysis (PCA) [Eyring et al., 2022] or into the latent space of deterministic autoencoders [Huguet et al., 2022]. Despite these approaches providing insight into fate mapping and cell state evolution, they do not take into account the distributional properties of single-cell data, like discreteness and overdispersion. Moreover, the standard OT formulation for modelling continuous trajectories assumes Euclidean geometry in the representation space [Peyré and Cuturi, 2017]. This hinders the application of deep representation techniques where the latent space is endowed with non-linear geometries, as linear paths in the latent space might not necessarily represent optimal paths in the observation space nor reflect the geometry of the data manifold [Arvanitidis et al., 2021].

To mitigate the restrictions of the Euclidean space assumption and incorporate the distributional properties of single-cell data into OT-based trajectory inference, we propose inducing Euclidean geometry into the latent space of a Variational Autoencoder (VAE) with Negative Binomial (NB) likelihood [Lopez et al., 2018]. To this end, we assume that single-cell data lie on a statistical manifold defined by the parameters of the underlying NB distribution and that the decoder model represents a map of the latent codes to the continuous parameter space of the data likelihood. To obtain a "flat" latent representation, we constrain the pullback metric from the decoder to approximate a scaled identity matrix (Fig. 1). Given a flat representation of single-cells, we take advantage of the Conditional Flow Matching (CFM) model [Tong et al., 2023] to simulate cellular dynamics, an approach to learning single-cell evolution through experimental time recently presented in Tong et al. [2023].

In summary, we propose the following contributions:

1. We introduce a flattening regularisation technique for VAEs with NB likelihood based on information geometry.

2. We combine our flattening regularisation with Conditional Flow Matching (CFM) for enhanced vector field modelling.

3. We showcase the applicability of our approach to the field of temporal modelling for single-cell computational biology.

## 2 Related work

**OT for single-cell genomics** Discrete OT methods in single-cell transcriptomics have shown potential in multiple application contexts, such as trajectory inference [Schiebinger et al., 2019], spatial reconstruction [Moriel et al., 2021] and multi-modal alignment [Klein et al., 2023]. Recent efforts have resorted to neural OT to predict the effects of drug perturbations on cells [Bunne et al., 2023] and cell trajectories in an unbalanced setting [Eyring et al., 2022]. Our work revolves around modelling gene expression through time via dynamic OT [Peyré and Cuturi, 2017], a combination that was extensively explored in prior works [Tong et al., 2020]. More specifically, the presented approach is inspired by MIOFlow [Huguet et al., 2022], where the authors model continuous single-cell trajectories in a latent space regularised such that latent Euclidean distances approximate the geodesic distances in the PCA space of single-cell data. However, MIOFlow's latent space regularisation is not tailored to the distributional properties of scRNA-seq measurements, an aspect that we explicitly consider by learning trajectories on a NB manifold.

**Geometry-regularised autoencoders** Enforcing the geometric structure of the data in the latent space of autoencoders was proposed on multiple occasions. Arvanitidis et al. [2021] model optimal latent paths based on the geometry of the ambient space defined by both deterministic and Gaussian stochastic decoders. Our research follows the theoretical framework introduced by Arvanitidis et al. [2022], which extends modelling geodesic latent paths to VAEs with arbitrary likelihood. More specifically, we use the theoretical concepts elaborated by the authors to describirie the geometry of the latent space of a discrete VAE model with a NB likelihood as a function of the decoder's pullback metric. Our approach is also inspired by Chen et al. [2020], where the authors enforce flatness in the latent space of VAEs by pushing the pullback metric tensor towards the identity matrix. However,

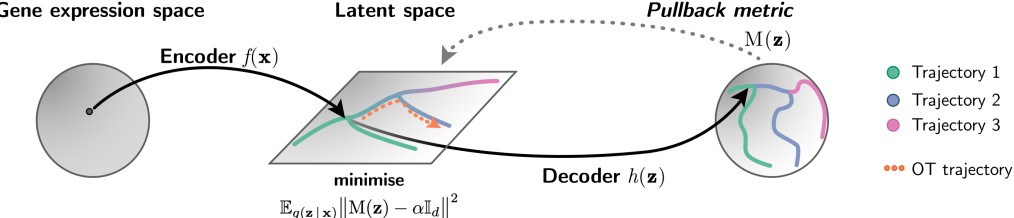

Figure 1: Visual conceptualisation of the proposed flattening approach for single-cell VAEs by regularising the pullback metric of a statistical manifold to match the identity matrix.

we additionally assume that the ambient space is a statistical manifold whose geometry is governed by the Fisher Information [Arvanitidis et al., 2022] rather than the Euclidean metric. Other relevant papers extend endowing the latent space with data geometry via isometric [Lee et al., 2022] and Jacobian [Nazari et al., 2023] regularisations.

**Flow-based OT**    Modelling continuous trajectories via Continuous Normalizing Flows (CNF) is a well-established practice in single-cell transcriptomics [Tong et al., 2020, Huguet et al., 2022]. Recently, novel approaches based on Flow Matching [Lipman et al., 2023] have yielded stabler and faster training by introducing simulation-free objectives to learn the CNF vector field for generative purposes [Liu et al., 2022, Pooladian et al., 2023, Albergo and Vanden-Eijnden, 2023]. Within such a context, significant efforts harnessed tractable simulation-free formulations of flows to address trajectory inference applications by drawing a connection with OT [Tong et al., 2023, Neklyudov et al., 2023]. Finally, our work relates to geometrically-informed Flow Matching, which was recently proposed by Chen and Lipman [2023].

## 3    Background

### 3.1    Modelling scRNA-seq with VAEs

The scRNA-seq method measures gene expression counts at the level of single cells. The discrete nature of observations readily allows modelling cell-specific counts using discrete likelihood models, inferring the continuous parameters of the data distribution by neural networks. More specifically, the biological and technical variability in the measurements lead to an inherent overdispersion of the expression counts, making the Negative Binomial (NB) likelihood the natural choice for modelling gene expression profiles.

More formally, given a gene expression matrix $\mathbf{X} \in \mathbb{N}_0^{N \times G}$ with $N$ cells and $G$ genes, we assume that each cell $\mathbf{x}$ is the realisation of a discrete random variable with the gene-specific distribution $\mathbf{X}_{ng} \sim \text{NB}(\boldsymbol{\mu}_{ng}, \boldsymbol{\theta}_g)$. Here, $\boldsymbol{\mu}$ and $\boldsymbol{\theta}$ represent the cell-gene-specific mean and the gene-specific inverse dispersion parameters, respectively. Further, we consider a latent variable model with marginalised density

$$p(\mathbf{x}) = \int p(\mathbf{x} \mid \mathbf{z}) p(\mathbf{z}) \mathrm{d}\mathbf{z} \,, \tag{1}$$

where $\mathbf{z}$ is a $d$-dimensional latent random variable with $d < G$ and $\mathbf{z} \sim \mathcal{N}(\mathbf{0}, \mathbb{I}_d)$ . In practice, we use amortised deep variational inference to jointly model the mean parameter of the data likelihood and the latent variables. This is achieved by utilising non-linear encoder and decoder functions of the form

$$\mathbf{z} = f_\psi(\mathbf{x}) \,, \tag{2}$$

$$\boldsymbol{\mu} = h_\phi(\mathbf{z}, l) = l \times \text{softmax}(\rho_\phi(\mathbf{z})) \,, \tag{3}$$

where $\rho_\phi : \mathbb{R}^d \to \mathbb{R}^G$ is a neural network that models the expression proportions of each gene in a cell and $l$ is the cell-specific size factor, which is directly derived from the data and refers to a cell's total number of counts $l = \sum_{g=1}^G \mathbf{x}_g$. The encoder formulation based on $f_\psi$ already takes into account the reparameterisation trick. In the following sections, we drop the dependency on the neural network parameters $\psi$ and $\phi$ for notational simplicity.

The gene-specific inverse dispersion $\boldsymbol{\theta}_g$ is considered a model parameter and trained via maximum likelihood together with the mean decoder $h$. The model is trained within the VAE framework, maximising the ELBO:

$$\mathcal{L}_{\text{ELBO}} = \mathbb{E}_{q(\mathbf{z}\,|\,\mathbf{x})}\big[\log p(\mathbf{x}\,|\,\mathbf{z};\boldsymbol{\theta})\big] - \text{KL}\big(q(\mathbf{z}\,|\,\mathbf{x})\|p(\mathbf{z})\big)\,. \tag{4}$$

As the representation learnt by $f$ is dense and compressed, modelling single-cell trajectories can be performed in the latent space of a discrete VAE. The low dimensionality coupled with continuity and the availability of a parametrised likelihood model makes $\mathbf{z}$ a natural proxy for studying dynamics in high-dimensional discrete data. Here, we couple this idea with continuous OT.

## 3.2 Dynamic OT

OT computes the most efficient mapping for transporting mass from one measure to another according to a pre-defined cost. Let $\nu$ and $\eta$ be marginal probability distributions defined on the spaces $\mathcal{X}$ and $\mathcal{Y}$, such that $\mathcal{X} = \mathcal{Y} = \mathbb{R}^d$. Moreover, consider the set of joint distributions $\Pi(\nu, \eta)$, such that for $\pi \in \Pi$ both $\pi(\mathrm{d}\mathbf{x}, \mathcal{Y}) = \nu$ and $\pi(\mathcal{X}, \mathrm{d}\mathbf{y}) = \eta$ holds. Considering the Euclidean distance cost $d(\mathbf{x}, \mathbf{y}) = \|\mathbf{x} - \mathbf{y}\|_2$ for observations $\mathbf{x} \in \mathcal{X}$ and $\mathbf{y} \in \mathcal{Y}$, the optimal coupling $\pi^*$ minimises the squared Wasserstein distance:

$$\begin{aligned} W_2(\nu, \eta)^2 &= \inf_{\pi \in \Pi} \int_{\mathcal{X} \times \mathcal{Y}} d(\mathbf{x}, \mathbf{y})^2 \mathrm{d}\pi(\mathbf{x}, \mathbf{y}) \\ &= \int_{\mathcal{X} \times \mathcal{Y}} d(\mathbf{x}, \mathbf{y})^2 \mathrm{d}\pi^*(\mathbf{x}, \mathbf{y})\,. \end{aligned} \tag{5}$$

Benamou and Brenier [2000] introduced a *dynamic formulation* of the OT problem. In this setting, let $p_t$ be a time-varying density over $\mathbb{R}^d$ constrained by $p_0 = \nu$ and $p_1 = \eta$. For a smooth time-dependent vector field $u : [0, 1] \times \mathbb{R}^d \to \mathbb{R}^d$ that satisfies the continuity equation

$$\frac{\mathrm{d}p}{\mathrm{d}t} = -\nabla \cdot (p_t u_t)\,, \tag{6}$$

the continuous counterpart to Eq. (5) is

$$W_2(\nu, \eta)^2 = \inf_{p_t, u_t} \int_0^1 \int_{\mathbb{R}^d} \|u_t(\mathbf{x})\|^2 p_t(\mathbf{x})\,\mathrm{d}\mathbf{x}\,\mathrm{d}t\,, \tag{7}$$

where $u_t(\mathbf{x}) = u(t, \mathbf{x})$. The parameterised field $u_t$ is said to *generate* the probability path $p_t$ if the latter is the solution of the continuity equation Eq. (6).

The velocity field $u_t(\mathbf{x})$ is associated with an Ordinary Differential Equation (ODE) $\mathrm{d}\mathbf{x} = u_t(\mathbf{x})\mathrm{d}t$. The solution of the ODE is an integration map $\phi_t(\mathbf{x})$ transporting mass along $u$ over time to match a source with a target distribution, given the initial condition $\phi_0(\mathbf{x}) = \mathbf{x}$. We deal with the setting where the vector field $u_t(\mathbf{x})$ is approximated by a neural network $v_\xi(t, \mathbf{x})$. In practice, one obtains $p_t$ by pushing points from the initial distribution $p_0$ through the integration map $\phi_t$.

## 3.3 Conditional Flow Matching

Tong et al. [2023] demonstrated that by expressing the time marginals $p_t(\mathbf{x})$ as a mixture of the form $p_t(\mathbf{x}) = \int p_t(\mathbf{x}\,|\,\mathbf{c})q(\mathbf{c})\mathrm{d}\mathbf{c}$ conditioned on some latent variables $\mathbf{c}$, the marginal vector field $u_t(\mathbf{x})$ is related to the conditional vector field $u_t(\mathbf{x}\,|\,\mathbf{c})$ through

$$u_t(\mathbf{x}) = \mathbb{E}_{q(\mathbf{c})}\left[\frac{u_t(\mathbf{x}\,|\,\mathbf{c})\,p_t(\mathbf{x}\,|\,\mathbf{c})}{p_t(\mathbf{x})}\right]\,. \tag{8}$$

Further, regressing $v_\xi(t, \mathbf{x})$ against the conditional field $u_t(\mathbf{x}\,|\,\mathbf{c})$ is equivalent to approximating the marginal field $u_t(\mathbf{x})$, up to a constant independent of the parameter $\xi$. Here, we follow the OT-CFM variant from Tong et al. [2023]. That is, given source and target sets of observations $\mathbf{X}_0$ and $\mathbf{X}_1$, we define the field-conditioning variable $\mathbf{c} = (\mathbf{x}_0, \mathbf{x}_1)$ as pairs of samples re-sampled from the static optimal coupling $\mathbf{c} \sim q(\mathbf{c}) = \pi^*(\mathbf{X}_0, \mathbf{X}_1)$. Assuming Gaussian marginals $p_t$ and $\mathbf{x}_0$ and $\mathbf{x}_1$ to be connected by Gaussian flows, both $p_t(\mathbf{x}\,|\,\mathbf{c})$ and $u_t(\mathbf{x}\,|\,\mathbf{c})$ become tractable:

$$p_t(\mathbf{x}\,|\,\mathbf{c}) = \mathcal{N}(t\mathbf{x}_1 + (1 - t)\mathbf{x}_0, \sigma^2)\,, \tag{9}$$

$$u_t(\mathbf{x} \mid \mathbf{c}) = \mathbf{x}_1 - \mathbf{x}_0 \,, \tag{10}$$

with a pre-defined value of $\sigma^2$. Accordingly, the CFM loss is

$$\mathcal{L}_{\text{CFM}} = \mathbb{E}_{t,q(\mathbf{c}),p_t(\mathbf{x}|\mathbf{c})} \big[ \|v_\xi(t, \mathbf{x}) - u_t(\mathbf{x} \mid \mathbf{c})\|^2 \big] \,, \tag{11}$$

with $t \sim \mathcal{U}(0, 1)$. From Eq. (9), one can readily see that OT-CFM uses straight paths to optimise the conditional vector field. Note, however, that straight lines in the latent space of a VAE may not reflect geodesic paths in the discrete observation space, i.e. gene space. We tackle this challenge by applying CFM to the latent space of a VAE with enforced Euclidean geometry.

## 4 Flattened NB-VAE for OT-based trajectory inference

### 4.1 The geometry of AEs

A common assumption is that the data lies near a low-dimensional Riemannian manifold $\mathcal{M}_{\mathcal{X}}$ whose coordinates are represented by the latent representation $\mathcal{Z}$. The decoder of a deterministic AE model can be seen as an immersion $h : \mathbb{R}^d \to \mathbb{R}^G$ of a latent space $\mathcal{Z} = \mathbb{R}^d$ into an embedded Riemannian manifold $\mathcal{M}_{\mathcal{X}}$ with a geometry-defining metric tensor $\mathrm{M}$ [Arvanitidis et al., 2021]. A Riemannian manifold is a smooth manifold $\mathcal{M}_{\mathcal{X}}$ endowed with a Riemannian metric $\mathrm{M}(\mathbf{x})$ for $\mathbf{x} \in \mathcal{M}_{\mathcal{X}}$. $\mathrm{M}(\mathbf{x})$ is a positive-definite matrix that changes smoothly and defines a local inner product on the tangent space $\mathcal{T}_{\mathbf{x}}\mathcal{M}_{\mathcal{X}}$ as the equation $\langle \mathbf{u}, \mathbf{v} \rangle_{\mathrm{M}} = \mathbf{u}^{\mathbf{T}}\mathrm{M}(\mathbf{x})\mathbf{v}$, with $\mathbf{v}, \mathbf{u} \in \mathcal{T}_{\mathbf{x}}\mathcal{M}_{\mathcal{X}}$ [Arvanitidis et al., 2021].

In this setting, the geometry of the latent space is directly linked to that of observation space through the *pullback metric* [Arvanitidis et al., 2022]

$$\mathrm{M}(\mathbf{z}) = \mathbb{J}_h(\mathbf{z})^{\mathrm{T}}\mathrm{M}(\mathbf{x})\mathbb{J}_h(\mathbf{z}) \,, \tag{12}$$

where $\mathbf{x} = h(\mathbf{z})$ and $\mathbb{J}_h(\mathbf{z})$ is the Jacobian matrix of $h(\mathbf{z})$. As such, the latent space $\mathcal{Z}$ is a Riemannian manifold whose properties are defined based on the geometry of $\mathcal{M}_{\mathcal{X}}$. An example of such properties is the shortest distance between two latent codes $\mathbf{z}_1$ and $\mathbf{z}_2$, which is expressed as the length of the shortest connecting curve along the manifold

$$d_{\text{latent}}(\mathbf{z}_1, \mathbf{z}_2) = \inf_{\gamma(t)} \int_0^1 \sqrt{\dot{\gamma}(t)^{\mathrm{T}}\mathrm{M}(\gamma(t))\dot{\gamma}(t)}\mathrm{d}t \,, \tag{13}$$

$$\gamma(0) = \mathbf{z}_1, \ \gamma(1) = \mathbf{z}_2 \,,$$

where $\gamma(t) : [0, 1] \to \mathbb{R}^d$ is a curve in the latent space and $\dot{\gamma}(t)$ its derivative along the manifold. If we assume Euclidean geometry in the observation space, we obtain $\mathrm{M}(\mathbf{z}) = \mathbb{J}_h(\mathbf{z})^{\mathrm{T}}\mathbb{J}_h(\mathbf{z})$. This applies since $\mathcal{M}_{\mathcal{X}}$ is endowed with the identity metric $\mathbb{I}_G$.

In VAEs, the decoder function $h$ maps a latent code $\mathbf{z} \in \mathcal{Z}$ to the parameter configuration $\boldsymbol{\mu} \in \mathcal{H}$ of the data likelihood, where $\mathcal{H} = \mathbb{R}^G$ represents the parameter space. As such, the decoder image lies on a statistical manifold, which is the smooth manifold of a probability distribution. Such manifolds have a natural metric tensor called Fisher Information Metric (FIM) [Nielsen, 2018, Arvanitidis et al., 2022]. The FIM defines the local geometry of the statistical manifold and can be used to build the pullback metric for arbitrary decoders. For a statistical manifold $\mathcal{M}_{\mathcal{H}}$ with parameters $\boldsymbol{\mu} \in \mathcal{H}$, the FIM is formulated as

$$\mathrm{M}(\boldsymbol{\mu}) = \mathbb{E}_{p(\mathbf{x}|\boldsymbol{\mu})} \big[ \nabla_{\boldsymbol{\mu}} \log p(\mathbf{x} \mid \boldsymbol{\mu}) \nabla_{\boldsymbol{\mu}} \log p(\mathbf{x} \mid \boldsymbol{\mu})^{\mathrm{T}} \big] \,, \tag{14}$$

where the metric tensor $\mathrm{M}(\boldsymbol{\mu}) \in \mathbb{R}^{G \times G}$ [Arvanitidis et al., 2022]. Analogous to deterministic AEs, one can combine Eq. (14) and Eq. (12) to formulate the pullback metric for an arbitrary statistical manifold, with the difference that the pulled-back metric tensor is defined in $\mathcal{H}$. As a result, the latent space of a VAE is endowed with the pullback metric for a statistical manifold

$$\mathrm{M}(\mathbf{z}) = \mathbb{J}_h(\mathbf{z})^{\mathrm{T}}\mathrm{M}(\boldsymbol{\mu})\mathbb{J}_h(\mathbf{z}) \,, \tag{15}$$

where $\mathrm{M}(\mathbf{z}) \in \mathbb{R}^{d \times d}$. Note that the calculation of the FIM is specific for the chosen likelihood type and, as such, depends on initial assumptions on the data distribution. Pulling back the FIM of the statistical manifold of NB probability distributions to the latent space $\mathcal{Z}$ we obtain

$$\mathrm{M}(\mathbf{z}) = \sum_g \frac{\boldsymbol{\theta}_g}{h_g(\mathbf{z})(h_g(\mathbf{z}) + \boldsymbol{\theta}_g)} \nabla_{\mathbf{z}} h_g(\mathbf{z}) \otimes \nabla_{\mathbf{z}} h_g(\mathbf{z}) \,, \tag{16}$$

where $\otimes$ is the outer product of vectors (see Section A.1 for more details on the derivation).

## 4.2 Flattening loss for a NB-VAE

Our goal is to perform OT in a latent space using a notion of distance that reflects geodesics along the NB statistical manifold. However, the computation of the geodesic distances for pairs of points requires learning geodesic curves connecting observations minimising Eq. (13) via gradient descent [Arvanitidis et al., 2022]. This is not feasible for CFM with OT-CFM, since the transport matrix requires calculating distances between all pairs of points from batches drawn from the source and the target distributions. Nevertheless, if $\mathbf{M}(\mathbf{z}) = \alpha \mathbb{I}_d$ for some fixed parameter $\alpha$, then the shortest distance between each pair of points in $\mathcal{Z}$ is given by the straight line between them.

In order to achieve this, we use an additional regularisation term (as in Chen et al. [2022] for the pulled-back Euclidean metric) to the VAE with NB likelihood, defined as:

$$\mathcal{L}_{\text{flat}}(\alpha) = \mathbb{E}_{q(\mathbf{z}|\mathbf{x})} \big\| \mathbf{M}(\mathbf{z}) - \alpha \mathbb{I}_d \big\|^2 , \qquad (17)$$

with $\mathbf{M}(\mathbf{z})$ represented by Eq. (16). In other words, we enforce a flat geometry in the latent space of a VAE with NB likelihood, where straight-line distances approximate geodesic distances on the statistical manifold. The loss of the derived VAE, therefore, is:

$$\mathcal{L} = \mathcal{L}_{\text{ELBO}} + \lambda \mathcal{L}_{\text{flat}} , \qquad (18)$$

where $\lambda$ controls the strength of the flattening regularisation. In practice, we train flows post hoc on the latent space of the regularised VAE and derive the evolution of gene counts through time via sampling the likelihood model from the decoded parameters. The training procedure is described in Algorithm 1.

---

**Algorithm 1** Train Flat NB-VAE

---

**Input:** Data matrix $\mathbf{X} \in \mathbb{N}_0^{N \times G}$, batch size $B$, maximum iterations $n_{\max}$, encoder $f_\psi$, decoder $h_\phi$, flatness loss scale $\lambda$
**Output:** Trained encoder $f_\psi$, decoder $h_\phi$ and inverse dispersion parameter $\boldsymbol{\theta}$
    randomly initialise gene-wise inverse dispersion $\boldsymbol{\theta}$
    randomly initialise the identity matrix scale $\alpha$ as a trainable parameter
    **for** $i = 1$ **to** $n_{max}$ **do**
        sample batch $\mathbf{X}^b \leftarrow \{\mathbf{x}_1, ..., \mathbf{x}_B\}$ from $\mathbf{X}$
        $\mathbf{l}^b \leftarrow \texttt{compute\_size\_factor}(\mathbf{X}^b)$
        $\mathbf{Z}^b \leftarrow f_\psi(1 + \log \mathbf{X}^b)$
        $\boldsymbol{\mu} \leftarrow h_\phi(\mathbf{Z}^b, \mathbf{l}^b)$
        $\mathcal{L}_{\text{KL}} \leftarrow \texttt{compute\_kl\_loss}(\mathbf{Z}^b)$
        $\mathcal{L}_{\text{recon}} \leftarrow \texttt{compute\_nb\_likelihood}(\mathbf{X}^b, \boldsymbol{\mu}, \boldsymbol{\theta})$
        $\mathbf{M}(\mathbf{Z}^b) \leftarrow$ Eq. (16)
        $\mathcal{L}_{\text{flat}} \leftarrow \texttt{MSE}(\mathbf{M}(\mathbf{Z}^b), \alpha \mathbb{I}_d)$
        $\mathcal{L} = \mathcal{L}_{\text{recon}} + \mathcal{L}_{\text{KL}} + \lambda \mathcal{L}_{\text{flat}}$
        Update parameters via gradient descent
    **end for**

---

## 5 Experiments

**Baselines & setup** We compare the latent space of our flat single-cell VAE (Flat NB-VAE) with that of a standard VAE (NB-VAE) trained with a NB decoder [Lopez et al., 2018] as representations for continuous OT. Additionally, we evaluate latent OT on the embeddings produced by the Geodesic AE (Geodesic AE) described in Huguet et al. [2022]. The latter method is trained on log-normalized gene expression to better accommodate the lack of a discrete probabilistic decoder. All three approaches are used to derive embeddings of time-resolved gene expression datasets. Subsequently, we use the cell representations to train an OT-CFM model [Tong et al., 2023] and learn latent trajectories using unpaired batches of observations from consecutive time points. Experimental details are provided in the following sections and Section A.4.

**Data** We evaluated our method for latent trajectory inference on four time-resolved single-cell RNA-seq datasets: (i) **Embryoid body (EB)**, Moon et al. [2019] profile 18,203 differentiating human embryoid cells over five time points, generating four lineages. (ii) **Pancreatic endocrinogenesis**, Bastidas-Ponce et al. [2019] measure 16,206 cells, spanning embryonic days 14.5 to 15.5, revealing multipotent cell differentiation into endocrine and non-endocrine lineages. (iii) **Cytomegalovirus (CMV) infection dataset**, Hein and Weissman [2021] measure 12,919 fibroblasts infected by Cytomegalovirus (CMV) across seven time points, focusing on the top 2000 highly variable genes. (iv) **Reprogramming dataset**, Schiebinger et al. [2019] explore the reprogramming of mouse embryonic fibroblasts into induced pluripotent stem cells, comprising 165,892 cells across 39-time points and 7 cell states, emphasising 1479 highly variable genes.

## 5.1  Flat latent space properties

We explore the relationship between the scaling of the flatness constraint and the likelihood score of the decoder. As measures of regularisation strength, we employ the *Variance of the Riemannian metric* (VoR) and the *magnification factor* (MF) following Chen et al. [2020] and Lee et al. [2022]. Briefly, given a metric tensor $M(\mathbf{z})$, VoR quantifies the uniformity of the Riemannian metric across space by computing the distance between $M(\mathbf{z})$ and its expected value $\bar{M} = \mathbb{E}_{\mathbf{z} \sim P_{\mathcal{Z}}}[M(\mathbf{z})]$. A VoR of 0 indicates constant metric across $\mathcal{Z}$. MF is defined as $MF(\mathbf{z}) = \sqrt{\det \bar{M}(\mathbf{z})}$ and reflects the proximity of the metric tensor to the identity $\mathbb{I}_d$. More details are in Section A.5.1. Table 1 depicts the properties of the latent space of a Flat NB-VAE at increasing levels of regularisation. Noticeably, high values of $\lambda$ contribute to a flatter latent space, suggesting our regularisation successfully induces a constant and flat geometry in the VAE bottleneck together with a gradual increase in negative log-likelihood. In practice, a value of $\lambda = 1$ is chosen for all datasets but the MEF reprogramming setting, where $\lambda = 0.1$ is selected to avoid an excessive penalisation of the gene reconstruction.

Table 1: VoR, MF and Negative Log-Likelihood (NLL) as functions of the flattening regularising scale $\lambda$. Missing entries indicate that the value evaluated in such a setting diverged to infinity.

|  | EB | | | CMV inf. | | | MEF reprogr. | | |
| --- | --- | --- | --- | --- | --- | --- | --- | --- | --- |
|  | VoR | MF | NLL | VoR | MF | NLL | VoR | MF | NLL |
| NB-VAE | 72.14 | - | 495.12 | 28.47 | - | 903.97 | 120.18 | - | 509.00 |
| Flat NB-VAE $\lambda = 0.1$ | 62.31 | 19231.44 | 519.46 | 21.29 | 3553.69 | 1006.15 | 12.89 | 1412.66 | 544.34 |
| Flat NB-VAE $\lambda = 1$ | 48.80 | 324.12 | 525.21 | 6.44 | 52.88 | 1072.15 | 21.45 | 120.46 | 553.01 |
| Flat NB-VAE $\lambda = 10$ | 1.23 | 4.17 | 576.91 | 1.72 | 4.83 | 1206.93 | 14.46 | 20.47 | 576.57 |

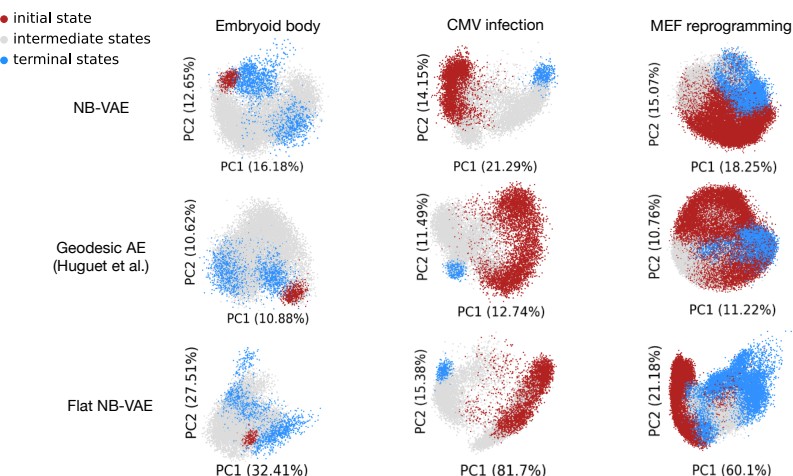

Figure 2: 2D PCA plots of the latent spaces computed by the Geodesic AE model, the NB-VAE and the Flat NB-VAE. Highlighted are initial, intermediate and terminal cell states along the biological trajectory.

We qualitatively compare the PCA embeddings of our Flat NB-VAE's latent space with competing models in Figure 2. The significantly higher variance explained by the first two principal components demonstrates that our flattening regularisation induces a lower latent space dimensionality, making the information compression more efficient and less noisy. At the same time, the biological structure of the data is preserved or even enhanced compared to competing methods. Biological preservation is particularly evident in the MEF dataset [Schiebinger et al., 2019], where our Flat NB-VAE induces a clearer separation between initial and terminal states, suggesting a better identification of the cellular dynamics.

## 5.2    Simulation of single-cell RNA-seq counts with OT-CFM

Coupled with OT-CFM, we compare simulating latent trajectories from NB-VAE with our Flat NB-VAE. Coherently with expectations, modelling latent trajectories with Euclidean cost benefits from a flat statistical manifold, as displayed by a lower 2-Wasserstein distance (WD) between real and simulated latent observations in Figure 3a. Aside from latent reconstruction, our approach allows us to gain insight into discrete gene expression counts and study how lineage drivers evolve through time (see Figure 3b for examples on the cardiac and neural crest branches of the EB dataset).

To quantify the performance of dynamic OT on our flat manifold, we adopt a similar setting to Tong et al. [2020]. More in detail, for each dataset, we leave out intermediate time points and train OT-CFM on the remaining cells. The capacity of OT to reconstruct unseen time point $t$ from $t-1$ during inference is an indication of the interpolation abilities of the model across the data manifold. Here, we use such a paradigm to compare different representation spaces. As a baseline, we calculate the distribution distance between the held-out time point $t$ and the ground truth cells at the previous time point $t-1$. In Table 2, we report the 2-Wasserstein distance, the polynomial-kernel Maximum Mean Discrepancy (MMD) and the mean $L^2$ distance between true

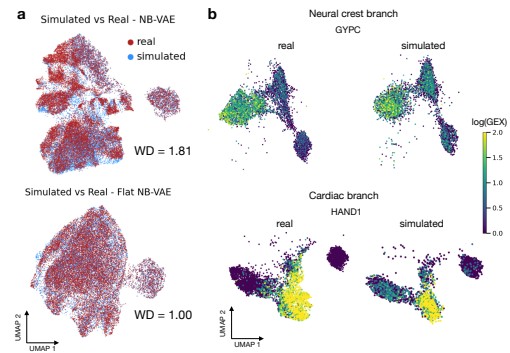

Figure 3: Simulating single-cell RNA-seq in time. (**a**) Overlap between real and simulated latent samples in the EB dataset. (**b**) Real and simulated cells on the flat manifold from the cardiac and neural crest lineages of the EB dataset. Colours indicate the $\log$ gene expression of the reported lineage drivers GYPC and HAND1.

and reconstructed latent cells. Furthermore, we add two measures in the decoded space, called *Density and Coverage* (D & C) [Naeem et al., 2020], which evaluate the mixing of real and simulated cell distributions in the gene expression space (see the Section A.5.1). On almost all datasets and metrics, trajectories in the flat latent space yield better overall latent time point reconstruction results. This observation is specifically true in the larger MEF reprogramming and Embryoid Body datasets. Furthermore, the experiments show that our approach yields an overall improvement in the inferred decoded trajectories, which can be seen by higher Density and Coverage metrics across all evaluated datasets.

Table 2: Comparison of held-out time point reconstruction across models, including the baseline.

| | EB | | | | | CMV inf. | | | | | MEF reprogr. | | | | |
| | Latent | | | Decoded | | Latent | | | Decoded | | Latent | | | Decoded | |
| | WD($\downarrow$) | MMD($\downarrow$) | $L^2$($\downarrow$) | $D$($\uparrow$) | $C$($\uparrow$) | WD($\downarrow$) | MMD($\downarrow$) | $L^2$($\downarrow$) | $D$($\uparrow$) | $C$($\uparrow$) | WD($\downarrow$) | MMD($\downarrow$) | $L^2$($\downarrow$) | $D$($\uparrow$) | $C$($\uparrow$) |
|---|---|---|---|---|---|---|---|---|---|---|---|---|---|---|---|
| Baseline | 2.87 | 0.56 | 0.57 | 0.04 | 0.13 | 2.77 | 0.52 | 0.55 | 0.25 | 0.18 | 3.21 | 0.83 | 0.83 | 0.12 | 0.07 |
| Geodesic AE | 2.15 | 0.36 | 0.40 | 0.01 | 0.02 | 2.09 | 0.32 | 0.34 | 0.02 | 0.03 | 2.49 | 0.54 | 0.57 | 0.01 | 0.00 |
| NB-VAE | 2.06 | 0.29 | 0.30 | 0.21 | 0.37 | 2.08 | **0.29** | **0.33** | 1.64 | 0.63 | 2.08 | 0.38 | 0.40 | 0.13 | 0.10 |
| Flat NB-VAE | **1.54** | **0.27** | **0.27** | **0.31** | **0.49** | **1.97** | 0.36 | 0.37 | **3.91** | **0.84** | **1.64** | **0.35** | **0.36** | **0.16** | **0.13** |

## 5.3 Latent vector field quality and lineage mapping

We hypothesise that dynamic OT with Euclidean cost benefits from being applied to a flat representation space, as the straight path between matched distributions inherently preserves the geometry enforced in the manifold. We evaluate the quality of the cell-specific directionality learned by our model using the Pancreas Endocrinogenesis dataset from Bastidas-Ponce et al. [2019]. More specifically, we follow the experimental setup introduced by Eyring et al. [2022], where a continuous vector field is learned by matching cell distributions through time. Using the Cellrank software [Lange et al., 2022, Weiler et al., 2023], we build random walks on a cell graph based on the directionality of latent velocities learned by OT-CFM. Walks converge to macrostates representing the end points of the biological process.

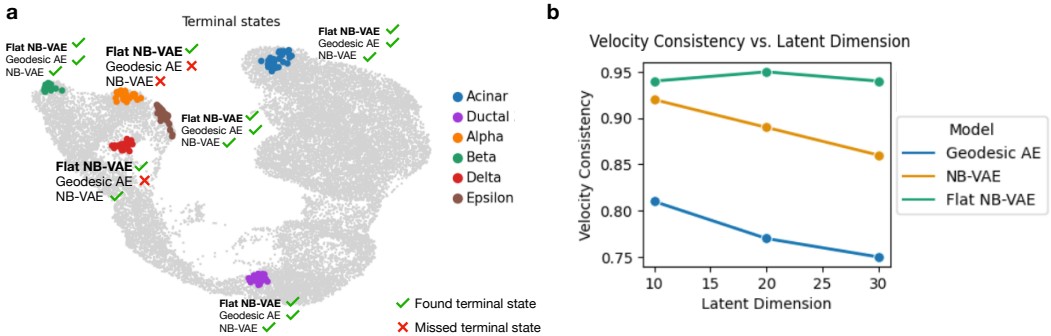

Figure 4: Learning terminal states from OT-CFM cell velocities. (**a**) Comparison of terminal states found by CellRank using 10-dimensional latent spaces. (**b**) Consistency computed for the latent velocities of cells across different latent space dimensionalities.

In Figure 4a, we summarise the number of terminal cell states identified by following the velocity graph. From prior biological knowledge, we know that the dataset contains six terminal states, all of which are identified in the latent space of our proposed Flat NB-VAE. Conversely, walks on the representations computed by the Geodesic AE and NB-VAE only capture four and five terminal states, respectively. In Figure 4b, we evaluate the velocity consistency within neighbourhoods of cells as a function of latent dimensionality. In contrast to the other methods, OT in our flat latent space yields a more consistent velocity field across latent dimensionalities.

## 6 Conclusion

We addressed the task of modelling temporal trajectories from unpaired cell distributions by using walks on flat NB manifolds. To achieve this, we proposed to regularise the pullback metric from the stochastic decoder of a single-cell VAE to approximate the identity matrix and enforce Euclidean geometry in the latent space. Our results demonstrate that our flattening procedure not only preserves but also enhances the biological structure in the latent space. By combining this approach with dynamic OT, we observed better prediction outcomes and more consistent vector fields on cellular manifolds. These improvements have practical benefits for central tasks in studying cellular development, such as fate mapping. In future work, we aim to extend the theory to include a broader range of statistical manifolds and single-cell tasks, such as modelling Poisson-distributed chromatin accessibility, estimating latent space distances for batch correction evaluation or enhancing OT-mediated perturbation modelling.

## Acknowledgments and Disclosure of Fundings

This work was supported by the German Federal Ministry of Education and Research (BMBF) (HOPARL, 031L0289A). AP and LH are supported by the Helmholtz Association under the joint research school "Munich School for Data Science - MUDS". FT consults for Immunai Inc., Singularity Bio B.V., CytoReason Ltd, and Omniscope Ltd, and has ownership interest in Dermagnostix GmbH and Cellarity.

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

## A   Appendix

### A.1   Derivation of the Fisher Information Metric for the Negative Binomial distribution

We first show that the Fisher Information of a univariate Negative Binomial (NB) distribution parameterised by mean $\mu$ and overdispersion $\theta$ with respect to $\mu$ is

$$\mathrm{M}(\mu) = \frac{\theta}{\mu(\mu + \theta)} \ . \tag{19}$$

We then move on with the derivation of the pullback metric in Eq. (16).

**Fisher Information of the NB distribution**   The univariate NB probability distribution parameterised by mean $\mu$ and inverse dispersion $\theta$ is

$$p_{\mathrm{NB}}(x \mid \mu, \theta) = \frac{\Gamma(\theta + x)}{x!\Gamma(\theta)} \Big(\frac{\theta}{\theta + \mu}\Big)^{\theta} \Big(\frac{\mu}{\theta + \mu}\Big)^{x} \ . \tag{20}$$

The Fisher Information of the distribution can be computed with respect to $\mu$ as:

$$\mathrm{M}(\mu) = -\mathbb{E}_{p(x|\mu,\theta)} \left[ \frac{\partial^2}{\partial \mu^2} \log p_{\mathrm{NB}}(x \mid \mu, \theta) \right] \ . \tag{21}$$

where

$$\log p_{\mathrm{NB}}(x \mid \mu, \theta) = C + \theta \left[\log(\theta) - \log(\theta + \mu)\right] + x \left[\log(\mu) - \log(\theta + \mu)\right] \ , \tag{22}$$

with $C = \log(\Gamma(\theta + x)) - \log(x!) - \log(\Gamma(\theta))$. Then, it can be shown that

$$\frac{\partial^2}{\partial \mu^2} \log p_{\mathrm{NB}}(x \mid \mu, \theta) = \frac{\theta + x}{(\theta + \mu)^2} - \frac{x}{\mu^2} \ . \tag{23}$$

Using the fact that the parameterisation involving the mean $\mu$ and inverse dispersion $\theta$ implies that

$$\mathbb{E}_{p(x|\mu,\theta)} \left[x\right] = \mu \ , \tag{24}$$

we can expand Eq. (21) as follows

$$\begin{aligned}
\mathrm{M}(\mu) &= -\mathbb{E}_{p(x|\mu,\theta)} \left[ \frac{\theta + x}{(\theta + \mu)^2} - \frac{x}{\mu^2} \right] \\
&= -\frac{1}{(\theta + \mu)^2} \mathbb{E}_{p(x|\mu,\theta)} \left[\theta + x\right] + \frac{1}{\mu^2} \mathbb{E}_{p(x|\mu,\theta)} \left[x\right] \\
&= \frac{\theta}{\mu(\mu + \theta)} \ .
\end{aligned} \tag{25}$$

**Derivation of the FIM**   We here consider the NB-VAE case, where the likelihood is parameterised by $\boldsymbol{\mu}_g = h_g(\mathbf{z})$ and $\boldsymbol{\theta}_g$ independently for each gene $g$.

When $h$ is continuously differentiable function of $\mathbf{z}$, the pullback metric $\mathrm{M}_g(\mathbf{z})$ of the output $g$ w.r.t $\mathbf{z}$ by the reparameterisation property [Lehmann and Casella, 1998] is

$$\begin{aligned}
\mathrm{M}_g(\mathbf{z}) &= \nabla_{\mathbf{z}} h_g(\mathbf{z}) \mathrm{M}(h_g(\mathbf{z})) \nabla_{\mathbf{z}} h_g(\mathbf{z})^T \\
&= \frac{\boldsymbol{\theta}_g}{h_g(\mathbf{z})(h_g(\mathbf{z}) + \boldsymbol{\theta}_g)} \nabla_{\mathbf{z}} h_g(\mathbf{z}) \otimes \nabla_{\mathbf{z}} h_g(\mathbf{z}) \ ,
\end{aligned} \tag{26}$$

where $\otimes$ is the outer product of vectors, and the gradients are column vectors.

By the chain rule, the joint Fisher Information of independent random variables equals the sum of the Fisher Information values of each variable [Zamir, 1998]. As all $\mathbf{x}_g$ are independent given $\mathbf{z}$ in the NB-VAE, the resulting Fisher Information Matrix is

$$\begin{aligned}
\mathrm{M}(\mathbf{z}) &= \sum_g \mathrm{M}_g(\mathbf{z}) \\
&= \sum_g \frac{\boldsymbol{\theta}_g}{h_g(\mathbf{z})(h_g(\mathbf{z}) + \boldsymbol{\theta}_g)} \nabla_{\mathbf{z}} h_g(\mathbf{z}) \otimes \nabla_{\mathbf{z}} h_g(\mathbf{z}) \ .
\end{aligned} \tag{27}$$

## A.2 The geometry of AEs

We deal with the assumption that the observed data lies near a Riemannian manifold $\mathcal{M}_\mathcal{X}$ embedded in the ambient space $\mathcal{X} = \mathbb{R}^G$. The manifold $\mathcal{M}_\mathcal{X}$ is defined as follows:

**Definition 1** *A Riemannian manifold is a smooth manifold $\mathcal{M}_\mathcal{X}$ endowed with a Riemannian metric* $\mathrm{M}(\mathbf{x})$ *for* $\mathbf{x} \in \mathcal{M}_\mathcal{X}$. $\mathrm{M}(\mathbf{x})$ *changes smoothly and identifies an inner product on the tangent space* $\mathcal{T}_\mathbf{x}\mathcal{M}_\mathcal{X}$ *at a point* $\mathbf{x} \in \mathcal{M}_\mathcal{X}$ *as* $\langle \mathbf{u}, \mathbf{v} \rangle_\mathrm{M} = \mathbf{u}^\mathbf{T}\mathrm{M}(\mathbf{x})\mathbf{v}$, *with* $\mathbf{v}, \mathbf{u} \in \mathcal{T}_\mathbf{x}\mathcal{M}_\mathcal{X}$.

For an embedded manifold $\mathcal{M}_\mathcal{X}$ with intrinsic dimension $d$, we can assume the existence of an invertible global chart map $\phi : \mathcal{M}_\mathcal{X} \to \mathbb{R}^d$ mapping the manifold $\mathcal{M}_\mathcal{X}$ to its intrinsic coordinates. A vector $\mathbf{v}_\mathbf{x} \in \mathcal{T}_\mathbf{x}\mathcal{M}_\mathcal{X}$ on the tangent space of $\mathcal{M}_\mathcal{X}$ can be expressed as a pushforward $\mathbf{v}_\mathbf{x} = \mathbb{J}_{\phi^{-1}}(\mathbf{z})\mathbf{v}_\mathbf{z}$ of a tangent vector $\mathbf{v}_\mathbf{z} \in \mathbb{R}^d$ at $\mathbf{z} = \phi(\mathbf{x})$, where $\mathbb{J}$ indicates the Jacobian. Therefore, $\mathbb{J}_{\phi^{-1}}$ maps vectors $\mathbf{v} \in \mathbb{R}^d$ into the tangent space of the embedded manifold $\mathcal{M}_\mathcal{X}$. Following Arvanitidis et al. [2021], the ambient metric $\mathrm{M}(\mathbf{x})$ can be represented in terms of intrinsic coordinates as:

$$\mathrm{M}(\mathbf{z}) = \mathbb{J}_{\phi^{-1}}(\mathbf{z})^T \mathrm{M}(\phi^{-1}(\mathbf{z}))\mathbb{J}_{\phi^{-1}}(\mathbf{z}) . \tag{28}$$

In other words, we can use the intrinsic formulation $\mathrm{M}(\mathbf{z})$ to compute quantities on the manifold, such as geodesic paths. However, for an embedded manifold $\mathcal{M}_\mathcal{X}$, the chart map $\phi$ is usually not known. A workaround is to define the geometry of $\mathcal{M}_\mathcal{X}$ on another Riemannian manifold $\mathcal{Z}$ with a trivial chart map $\phi(\mathbf{z}) = \mathbf{z}$, which can be mapped to $\mathcal{M}_\mathcal{X}$ via a smooth immersion $h$. In the next section, we elaborate on the connection between manifold learning and autoencoders following Arvanitidis et al. [2021, 2022].

### A.2.1 Deterministic AEs

We assume the decoder $h : \mathcal{Z} = \mathbb{R}^d \to \mathcal{X} = \mathbb{R}^G$ of a deterministic autoencoder is an immersion of the latent space into a Riemannian manifold $\mathcal{M}_\mathcal{X}$ embedded in $\mathcal{X}$ and with metric $\mathrm{M}$. This is valid if one also assumes that $d$ is the intrinsic dimension of $\mathcal{M}_\mathcal{X}$. As explained before, the Jacobian of the decoder maps tangent vectors $\mathbf{v}_\mathbf{z} \in \mathcal{T}_\mathbf{z}\mathcal{Z}$ in the latent space to tangent vectors $\mathbf{v}_{\mathbf{x}=h(\mathbf{z})} \in \mathcal{T}_\mathbf{x}\mathcal{M}_\mathcal{X}$. The decoder induces a metric into the latent space following Eq. (28) as

$$\mathrm{M}(\mathbf{z}) = \mathbb{J}_h(\mathbf{z})^T \mathrm{M}(h(\mathbf{z}))\mathbb{J}_h(\mathbf{z}) , \tag{29}$$

called *pullback metric*. The pullback metric defines the geometry of the latent space $\mathcal{Z}$ in relation to that of the manifold $\mathcal{M}_\mathcal{X}$. The metric tensor $\mathrm{M}(\mathbf{z})$ regulates the inner product of vectors $\mathbf{u}$ and $\mathbf{v}$ on the tangent space $\mathcal{T}_\mathbf{z}\mathcal{Z}$:

$$\langle \mathbf{u}, \mathbf{v} \rangle_\mathrm{M} = \mathbf{u}^T \mathrm{M}(\mathbf{z})\mathbf{v} . \tag{30}$$

To enhance latent representation learning, distances in the latent space $\mathcal{Z}$ can be optimised according to quantities of interest in the observation space $\mathcal{X}$, following the geometry of $\mathcal{M}_\mathcal{X}$. For instance, we can define the length of a curve $\gamma : [0, 1] \to \mathcal{Z}$ in the latent space by measuring its length on the manifold $\mathcal{M}_\mathcal{X} = h(\mathcal{Z})$:

$$\begin{aligned} L(\gamma) &= \int_0^1 \left\| \dot{h}(\gamma(t)) \right\| \mathrm{d}t \\ &= \int_0^1 \sqrt{\dot{\gamma}(t)^T \mathrm{M}(\gamma(t))\dot{\gamma}(t)} \mathrm{d}t , \end{aligned} \tag{31}$$

where the equality is derived by applying the chain rule of differentiation.

### A.2.2 Pulling back the information geometry

In machine learning, exploring latent spaces is crucial, especially in generative models like VAEs. One challenge is defining meaningful distances in the latent space $\mathcal{Z}$, which often depends on the properties of stochastic decoders and their alignment with the observation space $\mathcal{X}$. Injecting the geometry of the decoded space of a VAE into the latent space requires a different theoretical framework, where the data is assumed to lie near a statistical manifold.

VAEs can model different kinds of data types by using the decoder function as a non-linear likelihood parameter estimation model. We consider the decoder's output space as a parameter space $\mathcal{H}$ for a

probability density function. Depending on the data type, we express a likelihood function $p(\mathbf{x} \mid \boldsymbol{\theta})$ with parameters $\boldsymbol{\theta} \in \mathcal{H}$, reformulated as $p(\mathbf{x} \mid \mathbf{z})$ through a mapping $h : \mathcal{Z} \to \mathcal{H}$. We aim to define a natural distance measure in $\mathcal{Z}$ for infinitesimally close points $\mathbf{z}_1$ and $\mathbf{z}_2 = \mathbf{z}_1 + \delta\mathbf{z}$ when seen from $\mathcal{H}$. Arvanitidis et al. [2022] justify that such a distance corresponds to the Kullback-Leibler (KL) divergence:

$$\text{dist}^2(\mathbf{z}_1, \mathbf{z}_2) = \text{KL}(p(\mathbf{x} \mid \mathbf{z}_1), p(\mathbf{x} \mid \mathbf{z}_2)) . \tag{32}$$

To define the geometry of the statistical manifold, one can resort to Information Geometry, which studies probabilistic densities represented by parameters $\boldsymbol{\theta} \in \mathcal{H}$. In this framework, $\mathcal{H}$ becomes a statistical manifold equipped with a Fisher Information Metric (FIM):

$$\mathbf{M}(\boldsymbol{\theta}) = \int_{\mathcal{X}} [\nabla_{\boldsymbol{\theta}} \log p(\mathbf{x} \mid \boldsymbol{\theta})][\nabla_{\boldsymbol{\theta}} \log p(\mathbf{x} \mid \boldsymbol{\theta})]^T p(\mathbf{x} \mid \boldsymbol{\theta}) \, \mathrm{d}\mathbf{x} . \tag{33}$$

The FIM metric locally approximates the KL divergence. For the univariate case, it is known that

$$\text{KL}(p(x \mid \theta), p(x \mid \theta + \delta\theta)) \approx \frac{1}{2} \delta\theta^\top \mathbf{M}(\theta) \delta\theta + o(\delta\theta^2) . \tag{34}$$

In the VAE setting, we view the decoder not as a mapping to the observation space $\mathcal{X}$ but as a transformation to the parameter space $\mathcal{H}$. This perspective allows us to naturally incorporate the FIM metric into the latent space $\mathcal{Z}$. Consequently, the VAE can be seen as spanning a manifold $h(\mathcal{Z})$ in $\mathcal{H}$, with $\mathcal{Z}$ inheriting the metric in Eq. (33) via the Riemannian pullback. Based on this, we define a statistical manifold.

**Definition 2** *A statistical manifold is represented by a parameter space $\mathcal{H}$ of a distribution $p(\boldsymbol{x} \mid \boldsymbol{\theta})$ and is endowed with the FIM as the Riemannian metric.*

The Riemannian pullback metric is derived as in Eq. (29). Having defined the Riemannian pullback metric for VAEs with arbitrary likelihoods, one can extend the measurement of curve lengths in $\mathcal{Z}$ when mapped to $\mathcal{H}$ through $h$ as displayed by Eq. (31). This approach allows flexibility in the choice of the decoder, as long as the FIM of the chosen distribution type is tractable.

## A.3 Baseline description

Here, we describe the Geodesic Autoencoder (GAE) from Huguet et al. [2022]. For more details on the theoretical framework, we refer to the original publication. The GAE works by matching Euclidean distances between latent codes with the *diffusion geodesic distance*, which is an approximation of the diffusion ground distance in the observation space.

Briefly, the authors compute a graph with affinity matrix based on distances between observations $i$ and $j$ using a Gaussian kernel as:

$$(\mathbf{K}_\epsilon)_{ij} = k_\epsilon(\mathbf{x}_i, \mathbf{x}_j) , \tag{35}$$

with scale parameter $\epsilon$, where $\mathbf{x}_i, \mathbf{x}_j \in \mathcal{X}$ and $\mathcal{X}$ is the observation space. The affinity is then density-normalised by $\mathbf{M}_\epsilon = \mathbf{Q}^{-1} \mathbf{K}_\epsilon \mathbf{Q}^{-1}$, where $\mathbf{Q}$ is a diagonal matrix such that $\mathbf{Q}_{ii} = \sum_j (\mathbf{K}_\epsilon)_{ij}$. To compute the diffusion geodesic distance, the authors additionally calculate the diffusion matrix $\mathbf{P}_\epsilon = \mathbf{D}^{-1} \mathbf{M}_\epsilon$, with $\mathbf{D}_{ii} = \sum_{j=1}^n (\mathbf{M}_\epsilon)_{ij}$ and stationary distribution $\boldsymbol{\pi}_i = \mathbf{D}_{ii} / \sum_j \mathbf{D}_{jj}$. The diffusion geodesic distance between observations $\mathbf{x}_i$ and $\mathbf{x}_j$ is

$$G_\alpha(\mathbf{x}_i, \mathbf{x}_j) = \sum_{k=0}^K 2^{-(K-k)\alpha} \|(\mathbf{P}_\epsilon)_{i:}^{2^k} - (\mathbf{P}_\epsilon)_{j:}^{2^k}\|_1 + 2^{-(K+1)/2} \|\boldsymbol{\pi}_i - \boldsymbol{\pi}_j\|_1 , \tag{36}$$

with $\alpha \in (0, 1/2)$. The running value of $k$ in Eq. (36) defines the scales at which similarity between the random walks starting at $\mathbf{x}_i$ and $\mathbf{x}_j$ are computed.

Given the diffusion geodesic distance $G_\alpha$ defined in Eq. (36), the GAE model is trained such that the pairwise Euclidean distances between latent codes approximate the diffusion geodesic distances in the observation space $\mathcal{X}$, in a batch of size $B$. Given an encoder $f : \mathbb{R}^G \to \mathbb{R}^d$, the reconstruction loss is optimised alongside a geodesic loss

$$\mathcal{L}_{\text{geodesic}} = \frac{2}{B} \sum_{i=1}^N \sum_{j>i} (\|f(\mathbf{x}_i) - f(\mathbf{x}_j)\|_2 - G_\alpha(\mathbf{x}_i, \mathbf{x}_j))^2 . \tag{37}$$

Table 3: Hyperparameter sweeps for training Flat NB-VAE. The hidden dimension column excludes the latent space layer, which is set to 10 unless specified otherwise. In bold, is the selected value used to present the results in the main.

|  | batch size | hidden dims | $\lambda$ |
|---|---|---|---|
| EB | **32**, 256, 512 | [1024, 512, 256], [512, 256], **[256]** | 0.001, 0.01, 0.1, **1**, 10 |
| Pancreas | **32**, 256, 512 | [1024, 512, 256], [512, 256], **[256]** | 0.001, 0.01, 0.1, **1**, 10 |
| CMV | **32**, 256, 512 | [1024, 512, 256], [512, 256], **[256]** | 0.001, 0.01, 0.1, **1**, 10 |
| MEF | **32**, 256, 512 | [1024, 512, 256], [512, 256], **[256]** | 0.001, 0.01, **0.1**, 1, 10 |

## A.4 Model setup

**Experimental details for Autoencoder models** The Geodesic AE, NB-VAE and Flat NB-VAE models are trained via shallow 2-layer neural networks with hidden dimensions `[256, 10]`. Between consecutive layers, we include batch normalisation, as we found that it improves the reconstruction loss. Non-linearities are introduced by the `ELU` activation function. Models are trained for 1000 epochs monitoring the VAE loss for early stopping with a 20-epoch patience. The learning rate is set by default to `1e-3`. Additionally, we increase the KL divergence from 0 to 1 linearly across epochs in VAE models. The NB-VAE and Flat NB-VAE models were trained with a batch size of 32. For the geodesic AE model, we employed a batch size of 256 after sweeping all values in $\{64, 100, 256\}$. Importantly, the geodesic autoencoder was trained to reconstruct $\log$-normalised counts as opposed to the NB-VAE and Flat NB-VAE, since it does not assume a NB decoder. Finally, for training stability, all encoders were fed with $\log(1 + \mathbf{x})$ given an input $\mathbf{x}$. The list of hyperparameter sweeps for Flat NB-VAE together with the selected values based on the validation loss is provided in Table 3.

**Experimental details for OT-CFM** For OT-CFM we use a 3-layer MLP with 64 hidden units per layer, a `SELU` activation function and a learning rate of `1e-3`. The velocity network is fed with a latent state concatenated with a scalar representing the time used for interpolation. Following suggestions from the OT-CFM repository [2], in each epoch we collect batches of cells from all time points to compute the objective for backpropagation. The variance hyperparameter $\sigma$ is set to 0.1 by default.

## A.5 Evaluation metrics

### A.5.1 Metric description

**Magnification factor** The magnification factor [Bishop et al., 1997] is calculated as:

$$\text{MF}(\mathbf{z}) = \sqrt{\det \mathbf{M}(\mathbf{z})} \,. \tag{38}$$

Here, $\mathbf{M}(\mathbf{z})$ represents the Riemannian metric tensor. Proximity to 1 reflects the similarity of the metric to the identity matrix.

**Variance of the Riemannian metric** In assessing the Riemannian metric, we introduce a key evaluation called the Variance of the Riemannian Metric (VoR) [Pennec et al., 2006]. VoR is defined as the mean square distance between the Riemannian metric $\mathbf{M}(\mathbf{z})$ and its mean $\bar{\mathbf{M}} = \mathbb{E}_{\mathbf{z} \sim P_\mathcal{Z}}[\mathbf{M}(\mathbf{z})]$. As suggested in Lee et al. [2022], we compute the VoR employing an affine-invariant Riemannian distance metric $d$, expressed as:

$$d^2(A, B) = \sum_{i=1}^{m} \left( \log \lambda_i(B^{-1}A) \right)^2 \,, \tag{39}$$

where $\lambda_i(B^{-1}A)$ indicates the $i^{th}$ eigenvalue of the matrix $B^{-1}A$. VoR provides insights into how much the Riemannian metric varies spatially across different $\mathbf{z}$ values. When VoR is close to zero, it indicates that the metric remains constant throughout the support of $P_\mathbf{z}$. This evaluation procedure focuses solely on the spatial variability of the Riemannian metric and is an essential aspect of assessing the learned manifolds. Note that the expected value in Eq. (39) is estimated using latent batches of size 256.

---

[2]https://github.com/atong01/conditional-flow-matching

**Density** [Naeem et al., 2020] Let $\mathbf{Y}$ and $\mathbf{X}$ be sets of generated and real data points with $M$ and $N$ observations, respectively. The neighbourhood sphere $B(\mathbf{x}_i, \text{NND}_k(\mathbf{x}_i))$ is the spherical region around a real datapoint $\mathbf{x}_i$ with, as radius, the Nearest-Neighbourhood-Distance (NND), hence the distance between $\mathbf{x}_i$ the furthest of its k-nearest neighbours. For a generated sample $\mathbf{y}_i$, Density evaluates the number of real neighbourhood spheres that encompass $\mathbf{y}_j$. Mathematically, the metric is defined as:

$$\text{Density}(\mathbf{X}, \mathbf{Y}) = \frac{1}{kM} \sum_{j=1}^{M} \sum_{i=1}^{N} \mathbb{1}_{\mathbf{y}_j \in B(\mathbf{x}_i, \text{NND}_k(\mathbf{x}_i))} \ . \tag{40}$$

The Density metric rewards generated samples situated in regions where real samples are densely clustered. Importantly, Density can be higher than 1.

**Coverage** [Naeem et al., 2020] Similar to Density, Coverage builds a k-nearest-neighbourhood around the real samples as the sphere $B(\mathbf{x}_i, \text{NND}_k(\mathbf{x}_i))$. Given real and generated samples $\mathbf{X}$ and $\mathbf{Y}$ with $N$ and $M$ observations, Coverage is defined as:

$$\text{Coverage}(\mathbf{X}, \mathbf{Y}) = \frac{1}{N} \sum_{i=1}^{N} \left( \mathbb{1}_{\exists j \ \text{s.t.} \mathbf{y}_j \in B(\mathbf{x}_i, \text{NND}_k(\mathbf{x}_i))} \right) \ . \tag{41}$$

Thus, Coverage measures the fraction of real samples whose neighbourhoods contain at least one generated sample. The score is bounded between 0 and 1.

**Velocity consistency** [Gayoso et al., 2023] This metric quantifies the average Pearson correlation between the velocity $v(\mathbf{x}_j)$ of a reference cell $\mathbf{x}_j$ and the velocities of its neighbouring cells within the k-nearest-neighbour graph. It is mathematically expressed as:

$$c = \frac{1}{k} \sum_{\mathbf{x} \in \mathcal{N}_k(\mathbf{x}_j)} \text{corr}(v(\mathbf{x}_j), v(\mathbf{x})) \ . \tag{42}$$

Here $c$ represents the Velocity Consistency, $k$ denotes the number of nearest neighbours considered in the k-nearest-neighbour graph, $\mathbf{x}_j$ is the reference cell, $\mathcal{N}_k(\mathbf{x}_j)$ represents the set of neighbouring cells. The value $\text{corr}(v(\mathbf{x}_j), v(\mathbf{x}))$ is the Pearson correlation between the velocity of the reference cell $v(\mathbf{x}_j)$ and the velocity of each neighbouring cell $v(\mathbf{x})$. Higher values of $c$ indicate greater local consistency in velocity across the cell manifold.

### A.5.2 Metric computation

**Trajectory reconstruction experiments** In Table 2, we explore the performance of OT on different embeddings based on the reconstruction of held-out time points. For most datasets, we evaluate the leaveout performance on all intermediate time points. The only exception is the MEF reprogramming dataset [Schiebinger et al., 2019], where we conduct our evaluation holding out time points 2, 5, 10, 15, 20, 25 and 30 to limit the computational burden of the experiment. After training OT-CFM excluding the hold-out time point $t$, we collect the latent representations of cells at $t-1$ and simulate their trajectory until time $t$, where we compare the generated cells with the ground truth via distribution matching metrics. Density and Coverage in the gene expression space are used to evaluate the mixing between the real and generated cells. Their values are computed in the PCA space of 50 dimensions of the log-transformed real and generated gene expression, considering 10 neighbours. Finally, for the latent reconstruction quantification, generated latent cell distributions at time $t$ are standardised with the mean and standard deviation of the latent codes of real cells at time $t$ to make the results comparable across embedding models. Results in Table 2 are averaged across three seeds.

**Fate mapping with CellRank** Following the setting proposed by Eyring et al. [2022], we compute the latent velocity field with OT-CFM and input the velocities to CellRank [Lange et al., 2022, Weiler et al., 2023]. Using the function `g.compute_macrostates(n_states, cluster_key)` of the GPPC estimator for macrostate identification [Reuter et al., 2018], we look for 10 to 20 macrostates. If OT-CFM cannot find one of the 6 terminal states within 20 macrostates for a certain representation, we mark the terminal state as missed (see Figure 4). Terminal states are computed with the function `compute_terminal_states(method, n_states)`. Velocity consistency is estimated using the scVelo package [Bergen et al., 2020] through the function `scv.tl.velocity_confidence(adata_latent_flat)`. The value is then averaged across cells.

**Lineage driver analysis Figure 3b** We examine if lineage driver expression matches between real and predicted expression data. OT-based simulations are performed by pushing forward cells at time point $t_0$ using the learnt integration map. Pushed-forward cells are not annotated. We use label transfer to label cell clusters in the generated datasets based on the real data using the `sc.tl.ingest` Scanpy function [Wolf et al., 2018].

## A.6 Data

### A.6.1 Data description

**Embryoid Body (EB)** Moon et al. [2019] measured the expression of 18,203 differentiating human embryoid single cells across 5 time points. From an initial population of stem cells, approximately four lineages emerged, including Neural Crest, Mesoderm, Neuroectoderm and Endoderm cells. Here, we resort to a reduced feature space of 1241 highly variable genes. OT has been readily applied to the embryoid body datasets in multiple scenarios [Tong et al., 2020, 2023], making it a solid benchmark for time-resolved single-cell trajectory inference. The data is split into 80% training and 20% test sets.

**Pancreatic Endocrinogeneris (Pancreas)** We consider 16,206 cells from Bastidas-Ponce et al. [2019] measured across two time points corresponding to embryonic days 14.5 and 15.5. In the dataset, multipotent cells differentiate branching into endocrine and non-endocrine lineages until reaching 6 terminal states. Challenges concerning such dataset include bifurcation and unbalancedness of cell state distributions across time points [Eyring et al., 2022]. The data is split into 80% training and 20% test sets.

**Infection Dataset (CVM)** We include the single-cell dataset from Hein and Weissman [2021] which profiles 12,919 fibroblasts infected by Cytomegalovirus across 7 time points. For trajectory inference, we subset the dataset to include only the human genome with the 2000 top highly variable genes. A challenge with this dataset is its reduced number of observations compared with its counterpart examples. The data is split into 90% training and 10% test sets.

**Reprogramming Dataset (MEF)** We consider the dataset introduced in Schiebinger et al. [2019], which studies the reprogramming of Mouse Embryonic Fibroblasts (MEF) into induced Pluripotent Stem Cells (iPSC). The dataset consists of 165,892 cells profiled across 39 time points and 7 cell states. For this dataset, we keep 1479 highly variable genes. Due to its number of cells, such a dataset is the most complicated to model among the considered. The data was split into 80% training and 20% test sets.

### A.6.2 Data preprocessing

We use the Scanpy [Wolf et al., 2018] package for single-cell data preprocessing. The general pipeline involves normalisation via `sc.pp.normalize_total`, log-transformation via `sc.pp.log1p` and highly-variable gene selection using `sc.pp.highly_variable_genes`. 50-dimensional embeddings are then computed via PCA through `sc.pp.pca`. We then use the PCA representation to compute the 30-nearest-neighbour graphs around single observations and use them for learning 2D UMAP embeddings of the data. For the latter steps, we employ the scanpy functions `sc.pp.neighbours(adata)` and `sc.tl.umap(adata)`. For the CMV dataset [Hein and Weissman, 2021], we exclude the viral genome and only keep human gene expression.

## A.7 Details about computational resources

Our model is implemented in Python 3.10, and for deep learning models, we used PyTorch 2.0. For the implementation of NeuralODE-based simulations, we use the torchdyn package. Our experiments ran on different GPU servers with varying specifications:

- GPU: 16x Tesla V100 GPUs (32GB RAM per card)
- GPU: 2x Tesla V100 GPUs (16GB RAM per card)
- GPU: 8x A100-SXM4 GPUs (40GB RAM per card)

## A.8 Additional results

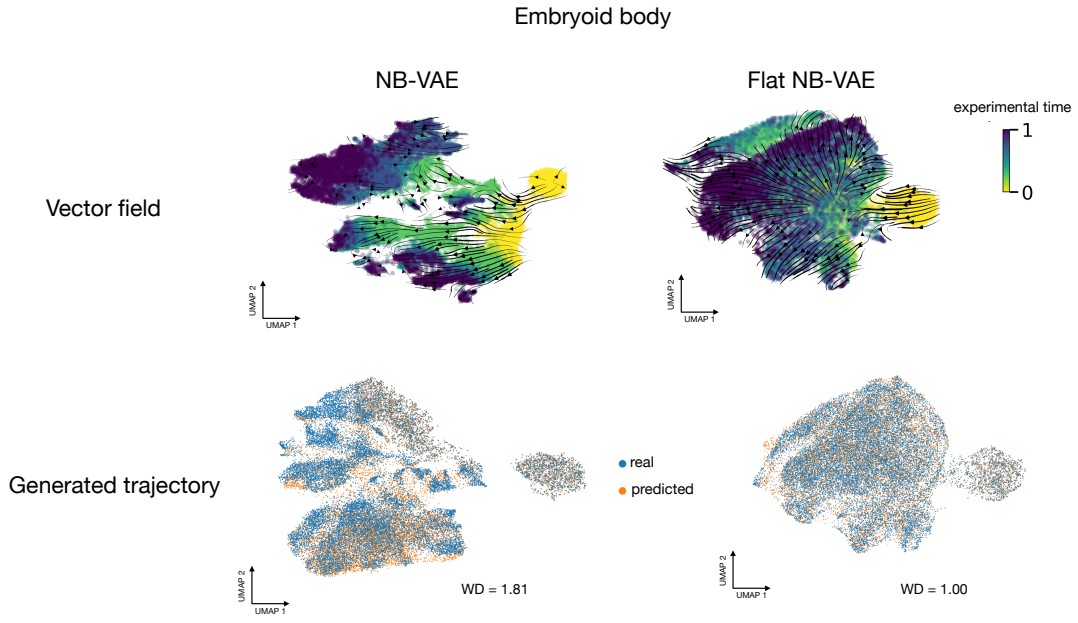

Figure 5: Latent vector field prediction visualised on 2D UMAP plots and latent reconstructions from OT-based simulations in the EB dataset. WD represents the 2-Wasserstein distance between real and generated latent representations of cells.

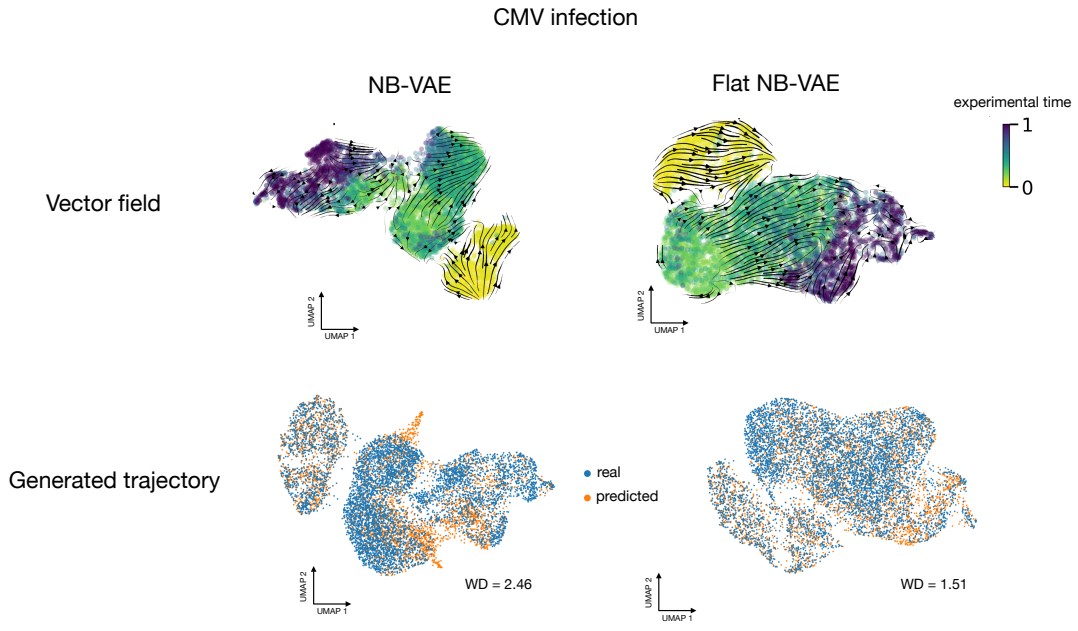

Figure 6: Latent vector field prediction visualised on 2D UMAP plots and latent reconstructions from OT-based simulations in the CMV infection dataset. WD represents the 2-Wasserstein distance between real and generated latent representations of cells.

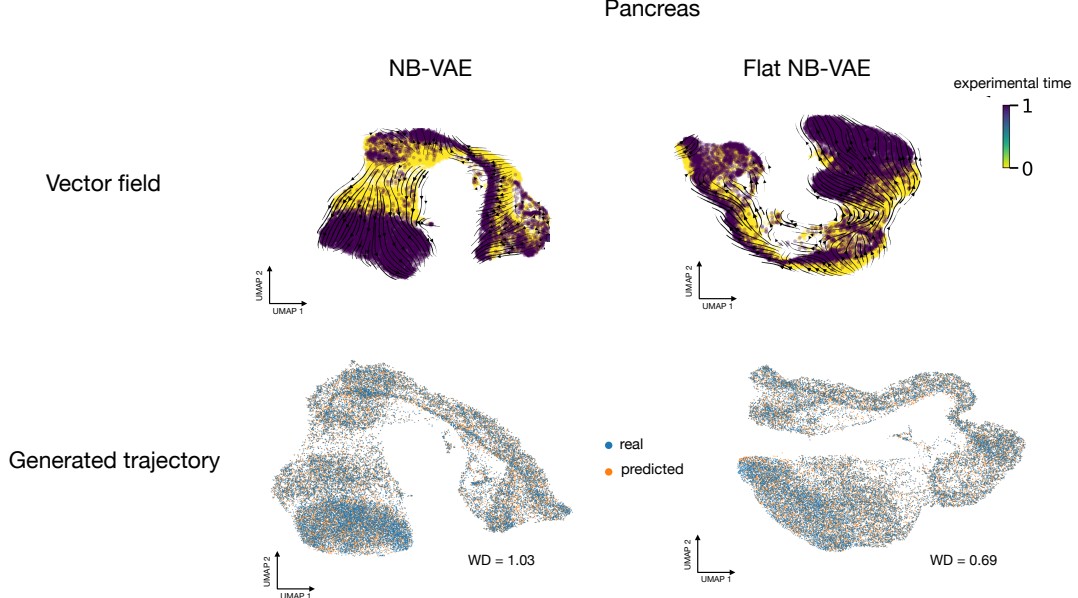

Figure 7: Latent vector field prediction visualised on 2D UMAP plots and latent reconstructions from OT-based simulations in the pancreas dataset. WD represents the Wasserstein distance between real and generated latent representations of cells.

