# OpenReview forum: "Modelling single-cell RNA-seq trajectories on a flat statistical manifold"
_NeurIPS.cc/2023/Workshop/AI4Science — NeurIPS2023-AI4Science Oral_

### Official Review · Reviewer_gqd8 · 2023-10-24
**Trajectory inference from time-course scRNA-seq data via conditional flow match optimal transport in the variational autoencoder latent space with a flattening regularizer**

**Rating:** 7
**Confidence:** 3

**Review:**

# Quality and Clarity
In this paper, the authors solve single-cell developmental trajectory inference problem from time-course single-cell data. The proposed method, based on optimal transport with conditional flow match in the latent space of a variational autoencoder. As the latent space is generally not Euclidean (Euclidean distances matches the geodesic distances), the authors introduce a regularizer to enforce Euclidean of the latent space.

The authors did a good job describing background, e.g., VAE, OT, geometry of the latent space, although the math derivatives in the appendix only gave the major results but missed detail steps.
Figure 1 is helpful, but miss information, e.g., time-course data. Also, I expected the right panel is a distorted version of the input, but the current panel is simply a larger version of the input.
Increasing $\lambda$ is expected to decrease VoR and increase the NLL, but this is not always true, as in Table 1 for the MEF data, where VoR increases when $\lambda = 10$ compared to $\lambda=0.1$. The authors did not discuss these results, e.g., stochasticity (the model was trained once), data size (an order more cells in this dataset compared to other datasets).
In figure 2, it's not very clear to me why it is a good thing for the first two PC to capture most of the variations, especially for complex dataset with confounding factors, e.g., experiment batches. If two factors are enough to capture the developmental trajectory, why not use a 2D latent space? Also, the initial and terminal state cells were in fact adjacent to each other in the PC space - likely all the three autoencoders did poorly based on the first two PCs.
Figure 4 axis was missing.

Generally, I think this is a good fit for the workshop but the quality of the paper can be improved but my rating is 7.


# Originality
The authors regularize VAE by enforcing a flat geometry in the latent space of the VAE, such that straight-line distances approximate geodesic distances.


# Significance of this work
Cellular trajectory inference from single-cell time-course data is an important problem. The proposed model may help solve this problem for biological discoveries.


# Pros
* The paper does a good job combines OT, CFM, VAE, to solve the trajectory inference problem from single-cell time-course data.
* The model shows promising results in benchmarking studies, especially for interpolation.
* Clearly written paper and the method seems to be sound

# Cons
* It's not clear enforcing a Euclidean latent space or using non-Euclidean geometry, e.g., hyperbolic geometry works better.
* Code and scripts for reproduce the results presented in the paper is not given.
* No discussion of time (used) complexity for different models thus not sure practitioners will adopt the model for data analysis.
* The pancreas dataset is a simple one and more complex data should be used to demonstrate the strength of the proposed approach.
* Some writing issue, i.e., NND (page 15, likely to be nearest neighbour distance) is not defined, similarly for N and M in define the density.

---

### Meta-Review · Area_Chair_xxrc · 2023-10-27

**Recommendation:** Accept (Oral)
**Confidence:** 4

**Metareview:**

In this paper, the author(s) present(s) Variational Autoencoder (VAE), using the flattening regularisation to ensure alignment between the latent space of a discrete Variational Autoencoder (VAE) and Euclidean space by reflecting geodesics in the negative binomial statistical manifold to solve the trajectory inference task on the single-cell RNA-seq data.

The paper is commendably well-organized, offering a comprehensive description of related methods and showcasing promising results in the provided experiments. However, the reviewer's valid concern revolves around the discussion of the regularization parameter $\lambda$, a critical element in the proposed model. Furthermore, I expect that conducting extensive experiments to evaluate the method's robustness in managing the inherent noise in scRNA-seq data would significantly enhance the paper.